# Tobacco smoking clusters in households affected by tuberculosis in an individual participant data meta-analysis of national tuberculosis prevalence surveys: Time for household-wide interventions?

Yohhei Hamada[1]*, Matteo Quartagno[2], Irwin Law[3], Farihah Malik[4], Frank Adae Bonsu[5], Ifedayo M. O. Adetifa[6,7], Yaw Adusi-Poku[5], Umberto D'Alessandro[6], Adedapo Olufemi Bashorun[6], Vikarunnessa Begum[8], Dina Bisara Lolong[9], Tsolmon Boldoo[10], Themba Dlamini[11], Simon Donkor[6], Bintari Dwihardiani[12], Saidi Egwaga[13], Muhammad N. Farid[14], Anna Marie Celina G. Garfin[15], Donna Mae G. Gaviola[15], Mohammad Mushtuq Husain[16], Farzana Ismail[17,18], Mugagga Kaggwa[19], Deus V. Kamara[13], Samuel Kasozi[20], Kruger Kaswaswa[21], Bruce Kirenga[22], Eveline Klinkenberg[23], Zuweina Kondo[13], Adebola Lawanson[24], David Macheque[25], Ivan Manhiça[25], Llang Bridget Maama-Maime[26], Sayoki Mfinanga[1,27,28,29], Sizulu Moyo[30,31], James Mpunga[21], Thuli Mthiyane[32], Dyah Erti Mustikawati[33], Lindiwe Mvusi[34], Hoa Binh Nguyen[35], Hai Viet Nguyen[35], Lamria Pangaribuan[33], Philip Patrobas[36], Mahmudur Rahman[16], Mahbubur Rahman[16], Mohammed Sayeedur Rahman[8], Thato Raleting[26], Pandu Riono[37], Nunurai Ruswa[38], Elizeus Rutebemberwa[39], Mugabe Frank Rwabinumi[22], Mbazi Senkoro[27], Ahmad Raihan Sharif[16], Welile Sikhondze[11], Charalambos Sismanidis[3], Tugsdelger Sovd[40], Turyahabwe Stavia[20], Sabera Sultana[8], Oster Suriani[33], Albertina Martha Thomas[38], Kristina Tobing[9], Martie Van der Walt[32], Simon Walusimbi[22], Mohammad Mostafa Zaman[8], Katherine Floyd[3], Andrew Copas[1], Ibrahim Abubakar[1], Molebogeng X. Rangaka[1,41]

1 Institute for Global Health, University College London, London, United Kingdom, 2 MRC Clinical Trials Unit, Institute of Clinical Trials and Methodology, University College London, London, United Kingdom, 3 Global Tuberculosis Programme, World Health Organization, Geneva, Switzerland, 4 UCL Great Ormond Street Institute of Child Health, University College London, London, United Kingdom, 5 National Tuberculosis Programme, Ghana Health Service, Accra, Ghana, 6 Disease Control and Elimination Theme, Medical Research Council Unit The Gambia at London School of Hygiene and Tropical Medicine, Banjul, Gambia, 7 Department of Infectious Diseases Epidemiology, London School of Hygiene & Tropical Medicine, London, United Kingdom, 8 World Health Organization, Country Office for Bangladesh, Dhaka, Bangladesh, 9 National Research and Innovation Agency, Jakarta, Indonesia, 10 Tuberculosis Surveillance and Research Department, National Center for Communicable Disease, Ulaanbaatar, Mongolia, 11 Eswatini National Tuberculosis Program, Ministry of Health, Mbabane, Eswatini, 12 Center for Tropical Medicine, Faculty of Medicine, Public Health and Nursing, Gadjah Mada University, Yogyakarta, Indonesia, 13 Tuberculosis and Leprosy Programme, Ministry of Health and Social Welfare, Dodoma, United Republic of Tanzania, 14 Expert TB Committee, Jakarta, Indonesia, 15 National TB Control Program, Department of Health, Manila, Philippines, 16 Institute of Epidemiology, Disease Control and Research (IEDCR), Dhaka, Bangladesh, 17 Centre for Tuberculosis: National Institute for Communicable Diseases, a Division of the National Health Laboratory Services, Johannesburg, South Africa, 18 Department of Medical Microbiology, University of Pretoria, Pretoria, South Africa, 19 World Health Organization, Country Office for Uganda, Kampala, Uganda, 20 National Tuberculosis Control Programme, Ministry of Health, Kampala, Uganda, 21 National Tuberculosis Programme, Ministry of Health, Lilongwe, Malawi, 22 Makerere University Lung Institute, Kampala, Uganda, 23 Department of Global Health, Amsterdam University Medical Centers, Amsterdam, the Netherlands, 24 National Tuberculosis and Leprosy Control Programme, Federal Ministry of Health, Abuja, Nigeria, 25 National Tuberculosis Program, Ministry of Health, Maputo, Mozambique, 26 National TB and Leprosy Programme, Ministry of Health, Maseru, Lesotho, 27 National Institute for Medical Research, Muhimbili Medical Research Centre, Dar es Salaam, United Republic of Tanzania, 28 Liverpool School of Tropical Medicine, Liverpool, United Kingdom, 29 Department of Epidemiology, Alliance for Africa Health and Research, Dar es Salaam, United Republic of Tanzania, 30 Human and Social



**Data Availability Statement:** The IPD database is housed in the UCL Data Repository. The IPD from

prevalence surveys that were utilized in our study falls under a specific data sharing agreement established with the original investigators. According to this agreement, these data are not to be distributed or made publicly available without explicit permission from these investigators. Requests to access the dataset can be addressed to igh.tb-ipd@ucl.ac.uk. The data will be shared upon approval from the authors of the original studies.

**Funding:** MQ is supported through the Medical Research Council unit grants MC_UU_00004/06 and MC_UU_00004/07. The funders had no role in study design, data collection and analysis, decision to publish, or preparation of the manuscript.

**Competing interests:** TSo declares a receipt of funding from the Global Fund for conducting the TB prevalence survey in Mongolia. All other authors declare no competing interests.

Capabilities Division, Human Sciences Research Council, Pretoria, South Africa, **31** School of Public Health and Family Medicine, University of Cape Town, Cape Town, South Africa, **32** South African Medical Research Council, Cape Town, South Africa, **33** Ministry of Health, Jakarta, Indonesia, **34** National Department of Health, Pretoria, South Africa, **35** National Tuberculosis Programme, Hanoi, Viet Nam, **36** World Health Organization, Country Office for Nigeria, Abuja, Nigeria, **37** Department of Biostatistics and Population, Faculty of Public Health, University of Indonesia, Depok, Indonesia, **38** National TB and Leprosy Programme, Ministry of Health and Social Services, Windhoek, Namibia, **39** Department of Health Policy, Planning and Management, Makerere University School of Public Health, Kampala, Uganda, **40** Department of Monitoring and Evaluation and Internal Audit, Ministry of Health, Ulaanbaatar, Mongolia, **41** Division of Epidemiology and Biostatistics & CIDRI-AFRICA, University of Cape Town, Cape Town, South Africa

* yohei.hamada0@gmail.com

## Abstract

Tuberculosis (TB) and non-communicable diseases (NCD) share predisposing risk factors. TB-associated NCD might cluster within households affected with TB requiring shared prevention and care strategies. We conducted an individual participant data meta-analysis of national TB prevalence surveys to determine whether NCD cluster in members of households with TB. We identified eligible surveys that reported at least one NCD or NCD risk factor through the archive maintained by the World Health Organization and searching in Medline and Embase from 1 January 2000 to 10 August 2021, which was updated on 23 March 2023. We compared the prevalence of NCD and their risk factors between people who do not have TB living in households with at least one person with TB (members of households with TB), and members of households without TB. We included 16 surveys (n = 740,815) from Asia and Africa. In a multivariable model adjusted for age and gender, the odds of smoking was higher among members of households with TB (adjusted odds ratio (aOR) 1.23; 95% CI: 1.11–1.38), compared with members of households without TB. The analysis did not find a significant difference in the prevalence of alcohol drinking, diabetes, hypertension, or BMI between members of households with and without TB. Studies evaluating household-wide interventions for smoking to reduce its dual impact on TB and NCD may be warranted. Systematically screening for NCD using objective diagnostic methods is needed to understand the actual burden of NCD and inform comprehensive interventions.

## Background

Low and middle-income countries (LMIC) face dual epidemics of tuberculosis (TB) and non-communicable diseases (NCD) [1]. The Global Burden of Disease estimates indicate that the number of people with NCD in LMIC increased from 3 billion to 3.5 billion between 2010 and 2019 [2]. NCD were responsible for around 60% of all deaths in 2019 in these countries [2]. Furthermore, TB mortality disproportionally affects these countries, with almost 99% of TB deaths in LMIC [3].

TB disease and some NCD have a bidirectional association. NCD, such as diabetes, enhance the risk of TB and worsen the outcomes of people with active TB [4,5]. For example, two systematic reviews reported a diabetes diagnosis increased the risk of active TB 1.5–3 folds [6,7]. Conversely, TB may adversely affect NCD risk [8]. Shared risk factors such as smoking, alcohol, and an unhealthy lifestyle likely drive the bidirectional association [1].

TB is associated with social mixing; people with TB and their close contacts may likely share risk factors for NCD and TB. This might cause a syndemic of chronic diseases in socially disadvantaged populations who are at risk of TB, and the syndemic can make them even more vulnerable to other diseases as shown in the COVID-19 pandemic. Contact-tracing for TB is an established intervention to identify additional people with TB, primarily targeting households [9]. Integration of NCD screening, care and prevention within this intervention might help reach household contacts at risk for TB and NCD. However, data on the clustering of NCD among household contacts are limited. Studies among the general population have reported NCD clustering within households [10,11]. However, few studies have examined the clustering of NCD and NCD risk factors in household contacts of people with TB. Shivakumar et al. reported in a study in India that nearly 40% of adult household contacts of people with TB had diabetes or pre-diabetes [12]. Another in South Africa reported that 17.4% of TB contacts had diabetes [13]. While both estimates were almost twice the national prevalence of diabetes, the lack of a control group precluded direct comparison with members of households without a known TB source within the same geographical area.

National TB prevalence surveys are population-based surveys whose primary aim is to estimate the national TB prevalence, conducted in countries with high TB burden [14]. Some prevalence surveys collected data on NCD and risk factors such as smoking and alcohol use. Since participants are invited per household, data from prevalence surveys allow us to examine the burden of those conditions in households affected by TB, compared to households without an individual diagnosed with TB.

We, therefore, conducted a systematic review and individual-participant data (IPD) meta-analysis of national TB prevalence surveys to understand if NCD and NCD risk factors cluster in members of households with TB.

## Methods

The protocol of this systematic review has been pre-registered (S1 Checklist). (https://www.crd.york.ac.uk/prospero/display_record.php?RecordID=272679).

### Search strategy and eligibility criteria

We included national and sub-national TB prevalence surveys in LMIC that reported at least one NCD or NCD risk factor (e.g. smoking, alcohol use) among participants.

National TB prevalence surveys enrol individuals ≥ 15 years old through random household sampling. Participants who have symptoms or chest X-ray findings suggestive of TB (or any lung abnormality, depending on each survey) submit sputum samples for testing by bacteriological tests such as smear, Gene Xpert, and/or culture as per the survey protocol. Detailed methodology is found elsewhere [14].

We identified eligible surveys using the list maintained by WHO. Additionally, we searched Medline (OVID) and Embase on 10 August 2021 to identify sub-national surveys published since 1 January 2000. The search was updated on 23 March 2023 to identify new surveys. The detailed search strategy is presented in S1 Appendix.

Two investigators independently reviewed titles and abstracts and then full-text articles in duplicate to identify eligible studies. Discrepancies were resolved through discussion.

### Outcome

The outcome was the prevalence of NCD or NCD risk factors (diabetes, hypertension, smoking, alcohol use, and body mass index [BMI]), based on the survey definition. Data on other

NCDs or risk factors were not available. To define TB cases, we used survey cases as defined in each survey, which were confirmed bacteriologically [14].

## Collection and pooling of IPD

We sought IPD from surveys found until 10 August 2021 (i.e. the initial search) to allow sufficient time for data cleaning, harmonisation, and analysis. For surveys found through the updated search, we sought aggregated data from published reports and by contacting the authors.

S1 Table presents a list of variables requested. We checked data for consistency with survey reports and potentially invalid and implausible values; we resolved queries by contacting the original investigators. For height, weight, and BMI, biologically implausible values were treated as missing, following criteria used in previous studies [15,16]. The frequency of alcohol drinking was classified differently by the surveys (S2 Table). The choice that maximised the available information was to classify alcohol drinking into three groups: drinking $\geq$ twice per week, once a week or less, vs no drinking.

## Quality assessment

Prevalence surveys were conducted following the WHO-recommended methodology, ensuring the representation of participants through random sampling and using recommended screening and diagnostic methods [14]. Further, there is no well-established tool to assess the quality of studies on the prevalence of diseases [17]. Thus, we assessed quality by focusing on items relevant to our analysis. We assessed the participation rate, methods for screening and diagnosis of active TB, methods for NCD diagnosis and checked missingness of outcome variables.

## Statistical analysis

### Handling of missing data

Not all outcome variables were collected in all surveys. We performed multiple imputation separately for each outcome by restricting to surveys that collected the outcome. To address sporadic missingness, we conducted multiple imputation using multilevel fully conditional specification (see S2 Appendix for details).

### Comparison of prevalence of NCD between members of households with TB and members of non-TB households

We presented the proportion of participants with NCD and NCD risk factors, stratified into three groups: people with TB, people who do not have TB in households with at least one person with TB (referred to as members of households with TB), and members of households without TB. We performed a multilevel logistic regression analysis to estimate the odds ratios for NCD and NCD risk factors, comparing members of households with TB and those without TB. For this analysis, alcohol drinking was dichotomised into $\geq$ twice per week vs $<$ twice per week, because a multinominal model failed to converge. The model included random intercepts for sampling clusters and fixed intercepts for surveys. Next, we examined the odds ratio for NCD and NCD risk factors, adjusting for age and gender of participants. The analysis did not intend to examine causal associations; instead, our main objective was simply to ascertain the overall increase in the prevalence of NCD/NCD risk factors in members of households with TB compared to those without TB who are of the same age and gender. While various

factors could contribute to this increase, they were not considered since any observed rise, irrespective of the causes, suggests a potential need for intervention.

The proportion of total variability due to between-study heterogeneity was quantified by calculating I-squared.

### Association between NCD and NCD risk factors of people with TB and those among household members

We examined if the NCD and NCD risk factors of people with TB was associated with the presence of NCD or NCD risk factors among their household members. We fitted a multilevel logistic regression model, including NCD/risk factor status, age, and gender of people with TB.

Next, we fitted the same model, including age and gender of the household members. When households included multiple people with TB, we randomly sampled one person with TB per household, and their characteristics were used in the models.

### Sensitivity analysis

First, we repeated the analysis using an alternative categorisation of alcohol drinking: any drinking vs no drinking. Second, we repeated the analyses by excluding countries that collected NCD data only among a subset of the participants. Third, we conducted a record-level quantitative bias analysis to explore the impact of the misclassification of diabetes and hypertension status (see details for S2 Appendix) [18].

Publication bias was not expected since WHO has a complete archive of national TB prevalence surveys.

### Patient consent statement

This IPD meta-analysis was approved by the University College London Research Ethics Committee (18969/001). All participants provided informed consent to participate in the primary surveys included in this meta-analysis.

## Results

### Search results and overall characteristics

From 21 eligible surveys found through the WHO archive, we received IPD from 16 surveys (n = 740,815) (Fig 1) [19–34]. The remaining five surveys (in Democratic People's Republic of Korea, Ethiopia, Myanmar, Rwanda, and Zimbabwe) did not provide data; reasons were not provided. We included 16 out of 21 eligible national surveys for smoking, nine out of 11 for diabetes (n = 427,922) [19–25,32,34], eight out of 10 for alcohol (n = 327,021) [20,22–25,30,32], four of four for BMI (n = 174,437) [20,23,24,26], and two of two for hypertension (n = 79,804) [20,23]. The database searches identified additional five eligible studies, all reporting only smoking status[35–39]; none of the studies responded to our request for IPD before the closure of data collection. All were sub-national surveys, including 286,340 participants and only collected data on smoking. The updated search did not find surveys that could be included in the meta-analysis (S1 Fig).

We included surveys conducted between 2012 and 2020, five in Asia and 11 in Africa (S3 Table). All participants were aged 15 years or older. The survey participation rate ranged from 56.8 to 90.9% (median: 77.2%) (S4 Table). In all surveys, there were fewer male participants than females (from 38.0 to 46.6%). In three surveys in which information was sought from all participants, the proportion of participants with DM ranged from 2.4 to 5.1%.

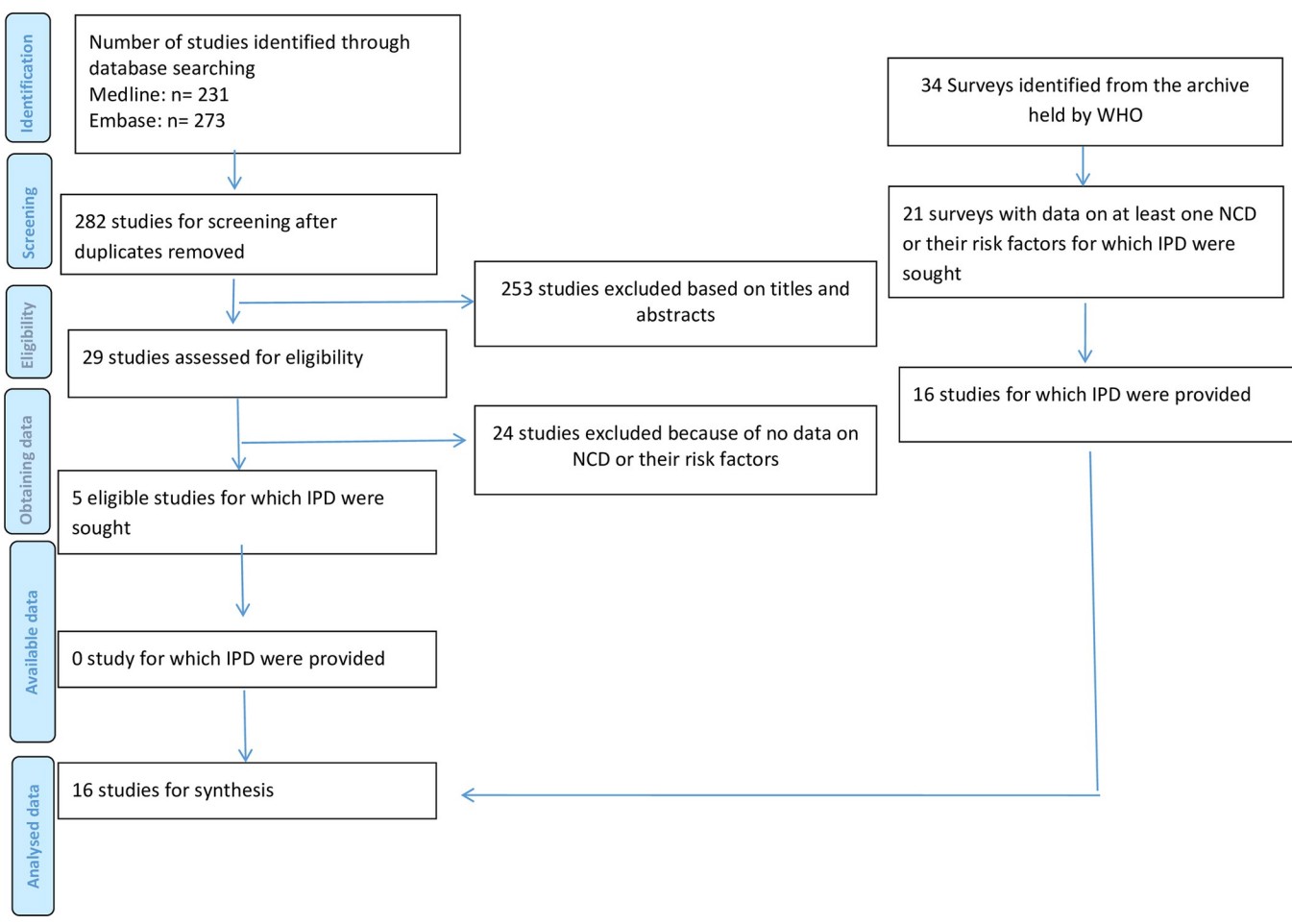

**Fig 1. Study selection.**

The median number of participants per household was two persons (interquartile range: 1–3). We did not find issues that could undermine IPD integrity.

In all surveys, diabetes was based on self-reports (S4 Table). For hypertension, one survey used a combination of blood pressure measurements and self-reports and the other used self-reports. Five surveys collected data on NCD and/or their risk factors from a subset of participants: participants eligible for sputum collection and a randomly selected subset of other participants in Eswatini, Namibia, and Mozambique [25,33], those eligible for sputum collection in the United Republic of Tanzania, and Viet Nam [24,34], and participants who had cough ≥ two weeks, had TB diagnosis, or treatment history in Ghana (S2 Fig and S4 Table) [22].

Table 1 presents the characteristics of participants. The median age was higher in people with TB at 44 years (interquartile range (IQR): 32–60) than in members of households with TB (median 34; IQR 22–50) and those without TB (median 35 years; IQR 24–50). A majority of people with TB were male (63.8%), while those in the other two groups were less likely to be male (40.0% in members of households with TB and 42.4% in those without TB). The diabetes prevalence was 5.6% in people with TB compared to 3.0% in members of households with TB and 3.2% in those without TB.

**Table 1. Demographic and clinical characteristics of study participants.**

| | Group | Members of households without TB | Members of households with TB | People with TB |
|---|---|---|---|---|
| Age | Median (IQR) years | 35 (24–50) | 34 (22–50) | 44 (32–60) |
| | N | 688767 | 7082 | 3245 |
| Gender | Male, n (%) | 291746 (42.4%) | 2831 (40.0%) | 2071 (63.8%) |
| | N | 688788 | 7082 | 3246 |
| Alcohol | No drinking, n (%) | 109961 (72.9%) | 1053 (68.5%) | 616 (58.6%) |
| | Weekly or less, n (%) | 35873 (23.8%) | 409 (26.6%) | 321 (30.5%) |
| | Twice per week or more, n (%) | 5026 (3.3%) | 75 (4.9%) | 115 (10.9%) |
| | N | 150860 | 1537 | 1052 |
| BMI | Mean (SD) kg/m$^2$ | 24.7 (5.1) | 24.3 (5.0) | 21.4 (4.4) |
| | N | 94000 | 1342 | 728 |
| Smoking | Current smoker | 95093 (19.5%) | 1218 (22.3%) | 1150 (36.8%) |
| | N | 487397 | 5471 | 3121 |
| Diabetes | Diabetes, n (%) | 6684 (3.2%) | 86 (3.0%) | 112 (5.6%) |
| | N | 209910 | 2875 | 2018 |
| HIV | Positive, n (%) | 16856 (10.6%) | 130 (9.1%) | 241 (21.2%) |
| | N | 159274 | 1424 | 1136 |
| Hypertension | Hypertension, n (%) | 20353 (36.2%) | 190 (33.4%) | 91 (25.6%) |
| | N | 56228 | 569 | 355 |

Note: Raw data before imputation. Denominators (N) vary by variables because of missing data.

BMI: body mass index; IQR: interquartile range; SD: standard deviation

## Prevalence of NCD and NCD risk factors in members of households with TB compared to members of households without TB

In the univariable model, members of households with TB were slightly more likely to smoke than members of households without TB (OR 1.12, 95% CI 1.05–1.20) (Table 2). When adjusting for age and gender, the odds of smoking were higher among members of households with TB (adjusted odds ratio (aOR) 1.23, 95% CI 1.11–1.38) compared with members of households without TB. The estimated aOR ranged from 0.87 to 1.78, with the highest observed in South Africa (I-squared = 35.4%, p = 0.15, tau$^2$ = 0.01) (S3 Fig). The higher prevalence of smoking among members of households with TB was observed both in Asian and African countries, and there was no significant difference by region (p = 0.0751) (S4 Fig). For alcohol drinking,

**Table 2. Prevalence of NCD/NCD risk factors in members of households with TB compared to those without TB.**

| | Current smoker | Alcohol drinking twice per week or more | Diabetes | Hypertension | BMI |
|---|---|---|---|---|---|
| Group | OR (95% CI) | OR (95% CI) | OR (95% CI) | OR (95% CI) | Difference (95%C) |
| Member of households without TB (Reference) | 1 | 1 | 1 | 1 | - |
| Members of households with TB (unadjusted) | 1.12 (1.05–1.20), p = 0.0013 | 1.18 (0.96–1.46), p = 0.1223 | 0.90 (0.74–1.10), p = 0.3013 | 0.91 (0.76–1.08), p = 0.2681 | -0.10 (-0.45; 0.25), p = 0. 5772 |
| Members of households with TB (Adjusted for age and gender) | 1.23 (1.11–1.38), p = 0.0003 | 1.19 (0.95–1.47), p = 0.1222 | 0.94 (0.77–1.15), p = 0.5333 | 0.88 (0.73–1.06), p = 0.1772 | -0.13 (-0.48; 0.21), p = 0.4431 |

Note: The estimates were from mixed-effects regression models accounting for clustering within surveys and sampling clusters.

OR: odds ratio; CI: confidence interval; BMI: body mass index

**Table 3. Association between NCD or their risk factors in people with TB and those in members of households with TB, adjusted for age and gender of TB patients.**

| NCD/NCD risk factors in people with TB in the same households | Current smoker<br>OR (95% CI) | Alcohol drinking twice per week or more<br>OR (95% CI) | Diabetes<br>OR (95% CI) | Hypertension<br>OR (95% CI) | BMI<br>Difference in Kg/m$^2$ (95% CI) |
|---|---|---|---|---|---|
| Current smoker | 1.47 (1.23–1.76), p < 0.0001 | | | | |
| Alcohol drinking twice per week or more | - | 1.45 (0.62–3.39), p = 0.3922 | - | - | - |
| Diabetes | - | - | 0.19 (0.00–77.81), p = 0.5782 | - | - |
| Hypertension | - | - | - | 1.31 (0.84–2.06), p = 0.2385 | - |
| BMI per 1 kg/m$^2$ increase | - | - | - | - | 0.08 (0.01–0.16), p = 0.0358 |

Note: Odds ratios were adjusted for age and gender of TB patients in the same households. Age and BMI were included in the model as continuous variables.

E.g. OR for age per 10-year increase indicates an increase in odds for each 10-year increase in age.

OR: odds ratio; CI: confidence interval; BMI: body mass index

there was no evidence that it was more common in members of households with TB (aOR 1.19; 95% CI 0.95–1.47) (Table 2 and S5 Fig).

We did not find evidence that the prevalence of diabetes or hypertension differed between members of households with and without TB (Table 2 and S6 and S7 Figs). Likewise, the mean BMI was not different between members of households with TB and those without TB (adjusted difference -0.13, 95% CI -0.48; 0.21) (Table 2 and S8 Fig).

## Predictors for NCD and NCD risk factors in members of households

In the models including NCD and NCD risk factors in people with TB, household members of people with TB who were current smokers were more likely to also be current smokers (aOR 1.46; 95% CI: 1.23–1.74) (Table 3). When age and gender of the household members were added to the models, current smoking of people with TB remained associated with their household members being current smokers (aOR 1.70; 95% CI 1.36–2.11, S5 Table). The proportion of variability due to between-study heterogeneity was small (I-squared = 0%, p = 0.54, tau$^2$ = 0.04), with aOR ranging from 0.99 to 4.27 (S9 Fig).

For alcohol drinking ($\geq$ twice per week vs less), hypertension and diabetes, the same conditions in people with TB did not significantly predict their presence in the household members (S5 Table and S10–S12 Figs). A higher BMI in people with TB was associated with higher BMI in their household members (Difference per 1kg/m$^2$ increase in BMI; 0.09 95% CI 0.02–0.16), but the level of the increase was small (S5 Table and S13 Fig).

## Sensitivity analysis

When alcohol drinking was dichotomised into any drinking vs no drinking (S6 Table), members of households with TB were significantly more likely to drink alcohol (aOR 1.19, 95% CI 1.06–1.33), while the point estimate did not differ from the primary analysis.

When excluding surveys that collected NCD data only in a subset of participants, the findings for smoking did not differ significantly (S7–S9 Tables). Alcohol drinking was significantly more common in members of households with TB (aOR 1.44, 95% CI 1.02–2.03) (S7 Table). S14 and S15 Figs present the sensitivity analysis exploring the impact of the misclassification of diabetes and hypertension. They suggest that the direction and magnitude of the association

vary significantly based on the accuracy of self-reported diagnosis, rendering the results inconclusive.

## Discussion

We included large nationally representative samples from both African and Asian countries with high TB incidence. The study suggests that smoking prevalence is higher in individuals living with people with TB, while the magnitude of the difference was small (aOR = 1.23, 95% CI 1.11–1.38). Participants were more likely to be current smokers if they lived with people with TB who were current smokers. To our knowledge, no study has investigated the clustering of smoking within households with people with TB. Clustering of smoking habits and of other conditions within households could increase the risk of TB development in household contacts.

Our model did not intend to elucidate causal associations and included only age and gender as covariates. The higher odds of smoking in household members of people with TB may reflect their tendency to share lifestyles or socioeconomic profiles. Previous studies reported the concordance of smoking among spousal pairs [10,11]. Since our model included only age and gender, the association is likely to be affected by other factors. Regardless of the mechanism explaining the association, a higher likelihood of smoking in members of households with TB suggests a need to address it, which is an important shared risk factor for TB and NCD.

The association between being members of households with TB and alcohol use was not significant in the primary analysis. In contrast, sensitivity analyses that changed the definition of alcohol use and another excluding six countries suggested an increased prevalence of alcohol use among them; because of the inconsistency, the association is inconclusive. A large proportion of missing data on alcohol drinking might have reduced the statistical power (despite imputation). Moreover, alcohol use could not be categorised clearly due to the variation in data collection.

In this study, diabetes and hypertension prevalence among members of households with TB did not differ from those without TB. However, the prevalence is likely underestimated because the ascertainment relied on participants' self-reports in most surveys. As shown in the sensitivity analysis, the true associations are inconclusive due to uncertainty in the level of under-detection. The national estimates of diabetes prevalence in survey countries range from 6% to 13% [3], while the prevalence in the present review ranged from 2.4 to 5.1% in countries. This suggests that a substantial number of household members of people with TB may not be aware of their diabetes. Diagnosing and treating diabetes can potentially reduce their risk of developing TB [40]; thus, screening contacts for diabetes as part of routine contact investigations could be reasonable alongside TB preventive treatment.

The strength of this study was the use of national survey data in 16 high TB burden countries from both Africa and Asia. While not all surveys provided data for all outcomes, each analysis included a large number of nationally representative participants. The study's major limitation is the use of self-reporting to ascertain NCD (i.e. diabetes and hypertension). This underestimates the NCD burden and might have biased the association. Future surveys should consider integrating standardised NCD data collection, such as the WHO STEPwise approach, to allow robust analysis of the interaction between NCD and people and households affected by TB [41]. Second, there was no data on other NCD, such as dyslipidaemia and chronic kidney disease. This underscores a need for a study examining the prevalence of various NCD, alone and as multimorbidity, through systematic screening using objective methods. Third, because of the small number of surveys for some outcomes, we used fixed slopes in our models.

Given that the associations are likely to be heterogeneous across countries, using random slopes would have been more appropriate.

## Conclusion

In prevalence surveys among the national representative populations, the self-reported prevalence of diabetes was lower than that expected from the national estimates, suggesting that diabetes was underdiagnosed. Data on other NCD were lacking, suggesting a missed opportunity to derive useful data from TB prevalence surveys. In contrast, the study found that smoking may be more frequent in members of households with TB, especially when index patients are smokers. Given the clustering of smoking in households with people with TB, household-level interventions addressing smoking may help reduce the risk for NCD and TB among household members. A well-designed prospective study applying systemic screening for NCD among contacts is needed to understand the accurate burden of NCD and risk factors.

## Supporting information

**S1 Checklist. PRISMA-IPD checklist of items to include when reporting a systematic review and meta-analysis of individual participant data (IPD).**
(DOCX)

**S1 Appendix. Search strategy.**
(DOCX)

**S2 Appendix. Supplementary methods.**
(DOCX)

**S1 Table. List of variables requested.**
(DOCX)

**S2 Table. Categorisations of current alcohol drinking by surveys.**
(DOCX)

**S3 Table. Characteristics of participants by survey.**
(DOCX)

**S4 Table. Quality of individual surveys.**
(DOCX)

**S5 Table. Association between NCD or their risk factors in people with TB and those in members of households with TB, adjusted for age and gender of TB patients and household members.**
(DOCX)

**S6 Table. Sensitivity analysis- the association between any alcohol drinking and TB status.**
(DOCX)

**S7 Table. Sensitivity analysis- prevalence of NCD/NCD risk factors in members of households with TB compared to those without TB.**
(DOCX)

**S8 Table. Sensitivity analysis- association between NCD or their risk factors in people with TB and those in members of households with TB, adjusted for age and gender of TB patients.**
(DOCX)

**S9 Table. Sensitivity analysis- association between NCD or their risk factors in people with TB and those in members of households with TB, adjusted for age and gender of TB patients and household members.**
(DOCX)

**S1 Fig. Results of the updated literature search.**
(DOCX)

**S2 Fig. Proportion of missing data by variable and survey.**
(DOCX)

**S3 Fig. Current smoking in members of households with TB compared to those without TB.**
(DOCX)

**S4 Fig. Current smoking in members of households with TB compared to those without TB, by region.**
(DOCX)

**S5 Fig. Alcohol drinking twice per week or more in members of households with TB compared to those without TB.**
(DOCX)

**S6 Fig. Diabetes in members of households with TB compared to those without TB.**
(DOCX)

**S7 Fig. Hypertension in members of households with TB compared to those without TB.**
(DOCX)

**S8 Fig. BMI in members of households with TB compared to those without TB.**
(DOCX)

**S9 Fig. Association between current smoking of people with TB and the same in their household members.**
(DOCX)

**S10 Fig. Association between alcohol drinking of people with TB and the same in their household members.**
(DOCX)

**S11 Fig. Association between diabetes of people with TB and the same in their household members.**
(DOCX)

**S12 Fig. Association between hypertension of people with TB and the same in their household members.**
(DOCX)

**S13 Fig. Association between BMI of people with TB and the same in their household members.**
(DOCX)

**S14 Fig. Sensitivity analysis-impact of misclassification of diabetes on its association with members of households with TB.**
(DOCX)

**S15 Fig. Sensitivity analysis-impact of misclassification of hypertension on its association with members of households with TB.**
(DOCX)

## Acknowledgments

We thank Dr.Craig Enders for his advice on multiple imputation. This paper is written in memory of Dr.Ayodele Awe, who worked as a National Professional Officer for Tuberculosis at WHO Country Office for Nigeria and supported the national TB prevalence survey.

## Author Contributions

**Conceptualization:** Yohhei Hamada, Matteo Quartagno, Farihah Malik, Andrew Copas, Ibrahim Abubakar, Molebogeng X. Rangaka.

**Data curation:** Yohhei Hamada, Irwin Law, Farihah Malik, Frank Adae Bonsu, Ifedayo M. O. Adetifa, Yaw Adusi-Poku, Umberto D'Alessandro, Adedapo Olufemi Bashorun, Vikarunnessa Begum, Dina Bisara Lolong, Tsolmon Boldoo, Themba Dlamini, Simon Donkor, Bintari Dwihardiani, Saidi Egwaga, Muhammad N. Farid, Anna Marie Celina G. Garfin, Donna Mae G. Gaviola, Mohammad Mushtuq Husain, Farzana Ismail, Mugagga Kaggwa, Deus V. Kamara, Samuel Kasozi, Kruger Kaswaswa, Bruce Kirenga, Eveline Klinkenberg, Zuweina Kondo, Adebola Lawanson, David Macheque, Ivan Manhiça, Llang Bridget Maama-Maime, Sayoki Mfinanga, Sizulu Moyo, James Mpunga, Thuli Mthiyane, Dyah Erti Mustikawati, Lindiwe Mvusi, Hoa Binh Nguyen, Hai Viet Nguyen, Lamria Pangaribuan, Philip Patrobas, Mahmudur Rahman, Mahbubur Rahman, Mohammed Sayeedur Rahman, Thato Raleting, Pandu Riono, Nunurai Ruswa, Elizeus Rutebemberwa, Mugabe Frank Rwabinumi, Mbazi Senkoro, Ahmad Raihan Sharif, Welile Sikhondze, Charalambos Sismanidis, Tugsdelger Sovd, Turyahabwe Stavia, Sabera Sultana, Oster Suriani, Albertina Martha Thomas, Kristina Tobing, Martie Van der Walt, Simon Walusimbi, Mohammad Mostafa Zaman, Katherine Floyd.

**Formal analysis:** Yohhei Hamada, Matteo Quartagno, Irwin Law, Andrew Copas.

**Investigation:** Farihah Malik, Frank Adae Bonsu, Ifedayo M. O. Adetifa, Yaw Adusi-Poku, Umberto D'Alessandro, Adedapo Olufemi Bashorun, Vikarunnessa Begum, Dina Bisara Lolong, Tsolmon Boldoo, Themba Dlamini, Simon Donkor, Bintari Dwihardiani, Saidi Egwaga, Muhammad N. Farid, Anna Marie Celina G. Garfin, Donna Mae G. Gaviola, Mohammad Mushtuq Husain, Farzana Ismail, Mugagga Kaggwa, Deus V. Kamara, Samuel Kasozi, Kruger Kaswaswa, Bruce Kirenga, Eveline Klinkenberg, Zuweina Kondo, Adebola Lawanson, David Macheque, Ivan Manhiça, Llang Bridget Maama-Maime, Sayoki Mfinanga, Sizulu Moyo, James Mpunga, Thuli Mthiyane, Dyah Erti Mustikawati, Lindiwe Mvusi, Hoa Binh Nguyen, Hai Viet Nguyen, Lamria Pangaribuan, Philip Patrobas, Mahmudur Rahman, Mahbubur Rahman, Mohammed Sayeedur Rahman, Thato Raleting, Pandu Riono, Nunurai Ruswa, Elizeus Rutebemberwa, Mugabe Frank Rwabinumi, Mbazi Senkoro, Ahmad Raihan Sharif, Welile Sikhondze, Charalambos Sismanidis, Tugsdelger Sovd, Turyahabwe Stavia, Sabera Sultana, Oster Suriani, Albertina Martha Thomas, Kristina Tobing, Martie Van der Walt, Simon Walusimbi, Mohammad Mostafa Zaman, Katherine Floyd.

**Methodology:** Yohhei Hamada, Matteo Quartagno, Irwin Law, Ibrahim Abubakar, Molebogeng X. Rangaka.

**Supervision:** Andrew Copas, Ibrahim Abubakar, Molebogeng X. Rangaka.

**Writing – original draft:** Yohhei Hamada.

**Writing – review & editing:** Yohhei Hamada, Matteo Quartagno, Irwin Law, Farihah Malik, Frank Adae Bonsu, Ifedayo M. O. Adetifa, Yaw Adusi-Poku, Umberto D'Alessandro, Adedapo Olufemi Bashorun, Vikarunnessa Begum, Dina Bisara Lolong, Tsolmon Boldoo, Themba Dlamini, Simon Donkor, Bintari Dwihardiani, Saidi Egwaga, Muhammad N. Farid, Anna Marie Celina G. Garfin, Donna Mae G. Gaviola, Mohammad Mushtuq Husain, Farzana Ismail, Mugagga Kaggwa, Deus V. Kamara, Samuel Kasozi, Kruger Kaswaswa, Bruce Kirenga, Eveline Klinkenberg, Zuweina Kondo, Adebola Lawanson, David Macheque, Ivan Manhiça, Llang Bridget Maama-Maime, Sayoki Mfinanga, Sizulu Moyo, James Mpunga, Thuli Mthiyane, Dyah Erti Mustikawati, Lindiwe Mvusi, Hoa Binh Nguyen, Hai Viet Nguyen, Lamria Pangaribuan, Philip Patrobas, Mahmudur Rahman, Mahbubur Rahman, Mohammed Sayeedur Rahman, Thato Raleting, Pandu Riono, Nunurai Ruswa, Elizeus Rutebemberwa, Mugabe Frank Rwabinumi, Mbazi Senkoro, Ahmad Raihan Sharif, Welile Sikhondze, Charalambos Sismanidis, Tugsdelger Sovd, Turyahabwe Stavia, Sabera Sultana, Oster Suriani, Albertina Martha Thomas, Kristina Tobing, Martie Van der Walt, Simon Walusimbi, Mohammad Mostafa Zaman, Katherine Floyd, Andrew Copas, Ibrahim Abubakar, Molebogeng X. Rangaka.

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
