## [Decision Letter · Decision Letter 0]

15 Dec 2023

PGPH-D-23-01887

Tobacco smoking clusters in households affected by tuberculosis in an individual participant data meta-analysis of national tuberculosis prevalence surveys: time for household-wide interventions?

Dear Dr. HAMADA,

Thank you for submitting your manuscript to PLOS Global Public Health. After careful consideration, we feel that it has merit but does not fully meet PLOS Global Public Health’s publication criteria as it currently stands. Therefore, we invite you to submit a revised version of the manuscript that addresses the points raised during the review process.

Please note that we have only been able to secure a single reviewer to assess your manuscript. We are issuing a decision on your manuscript at this point to prevent further delays in the evaluation of your manuscript. Please be aware that the editor who handles your revised manuscript might find it necessary to invite additional reviewers to assess this work once the revised manuscript is submitted. However, we will aim to proceed on the basis of this single review if possible. 

We look forward to receiving your revised manuscript.

Kind regards,

Jianhong Zhou

Staff Editor

Journal Requirements:

Additional Editor Comments (if provided):

Reviewers' comments:

Reviewer's Responses to Questions

**Comments to the Author**

1. Does this manuscript meet PLOS Global Public Health’s publication criteria? Is the manuscript technically sound, and do the data support the conclusions? The manuscript must describe methodologically and ethically rigorous research with conclusions that are appropriately drawn based on the data presented.

Reviewer #1: Yes

2. Has the statistical analysis been performed appropriately and rigorously?

Reviewer #1: Yes

3. Have the authors made all data underlying the findings in their manuscript fully available (please refer to the Data Availability Statement at the start of the manuscript PDF file)?

Reviewer #1: Yes

4. Is the manuscript presented in an intelligible fashion and written in standard English?

Reviewer #1: Yes

5. Review Comments to the Author

Reviewer #1: This study evaluates clustering of non-communicable diseases and risk factors among households of people with TB as compared to households without TB using data from TB prevalence surveys. The authors find that people from households affected by TB are more likely to be smokers. Of note is that surveys did not consistently collect data on NCD and risk factors and for some NCDs (hypertension, diabetes) self-reporting may have underestimated their prevalence. This highlights the importance of collecting data on NCD during TB prevalence surveys to better understand burden and improve referral and management for individuals in households affected by TB. The article is very well written, analysed and interpreted and makes a compelling case for integrating NCD data collection within surveys.

line 191 provide in the supplement (methods) additional details on how quality was assessed

were there any geographic/regional differences in smoking clustering

6. PLOS authors have the option to publish the peer review history of their article (what does this mean?). If published, this will include your full peer review and any attached files.

**Do you want your identity to be public for this peer review?** For information about this choice, including consent withdrawal, please see our Privacy Policy.

Reviewer #1: No

---

## [Decision Letter · Decision Letter 1]

29 Jan 2024

Tobacco smoking clusters in households affected by tuberculosis in an individual participant data meta-analysis of national tuberculosis prevalence surveys: time for household-wide interventions?

PGPH-D-23-01887R1

Dear Dr. HAMADA,

We are pleased to inform you that your manuscript 'Tobacco smoking clusters in households affected by tuberculosis in an individual participant data meta-analysis of national tuberculosis prevalence surveys: time for household-wide interventions?' has been provisionally accepted for publication in PLOS Global Public Health.

Best regards,

Palash Chandra Banik, MPhil

Academic Editor

Reviewer Comments (if any, and for reference):

Reviewer's Responses to Questions

**Comments to the Author**

1. If the authors have adequately addressed your comments raised in a previous round of review and you feel that this manuscript is now acceptable for publication, you may indicate that here to bypass the “Comments to the Author” section, enter your conflict of interest statement in the “Confidential to Editor” section, and submit your "Accept" recommendation.

Reviewer #1: All comments have been addressed

2. Does this manuscript meet PLOS Global Public Health’s publication criteria? Is the manuscript technically sound, and do the data support the conclusions? The manuscript must describe methodologically and ethically rigorous research with conclusions that are appropriately drawn based on the data presented.

Reviewer #1: Yes

3. Has the statistical analysis been performed appropriately and rigorously?

Reviewer #1: Yes

4. Have the authors made all data underlying the findings in their manuscript fully available (please refer to the Data Availability Statement at the start of the manuscript PDF file)?

Reviewer #1: Yes

5. Is the manuscript presented in an intelligible fashion and written in standard English?

Reviewer #1: Yes

6. Review Comments to the Author

Reviewer #1: The authors have addressed all my comments.

7. PLOS authors have the option to publish the peer review history of their article (what does this mean?). If published, this will include your full peer review and any attached files.

**Do you want your identity to be public for this peer review?** For information about this choice, including consent withdrawal, please see our Privacy Policy.

Reviewer #1: **Yes: **Ioana D Olaru
